# Preparation of Wear-Resistant Superhydrophobic Coatings Based on a Discrete-Phase Adhesive

Xuejuan Liu [1,2], Zhenxia Shi [1,2], Lin Lin [1,2], Xiaolan Shang [1,2], Jing Wang [1,2], Chunyan Xie [1,2,*] and Lei Wang [3,*]

1   College of Life Science, Langfang Normal University, Langfang 065000, China
2   Technical Innovation Center for Utilization of Edible and Medicinal Fungi in Hebei Province, Langfang 065000, China
3   Beijing Key Laboratory of Cryo-Biomedical Engineering, Technical Institute of Physics and Chemistry, Chinese Academy of Sciences, Beijing 100190, China
*   Correspondence: 1231597@lfnu.edu.cn (C.X.); leiwang@mail.ipc.ac.cn (L.W.)

**Abstract:** Among the many methods to prepare superhydrophobic coatings, the liquid spraying method has the advantages of simple operation, low equipment and substrate requirements, and a low cost to achieve large-scale industrialization. However, superhydrophobic coatings prepared using the existing one-step and two-step spraying methods are not wear resistant, and the failure mechanism is also not clear. After preparing coatings using existing methods and conducting wear tests, we show that the reason for their lack of wear resistance is the excessive bonding caused by the nanoparticles embedded in the continuous-phase adhesive, or the low bonding caused by adhesion to the adhesive surface. Based on the above conclusion, we propose a method to transform a continuous-phase adhesive into a discrete-phase adhesive via phase separation, after which it is mixed with nanoparticles for spraying. This new method allows the nanoparticles to bond to the adhesive while avoiding embedding, which avoids the shortcomings of existing methods. Consequently, coatings fabricated using the new method have better wear resistance properties and important significance for practical applications.

**Keywords:** wear-resistant; discrete-phase; superhydrophobic coatings

## 1. Introduction

A superhydrophobic surface has low solid–liquid adhesiveness [1] since the apparent contact angle of a liquid droplet on this type of surface exceeds 150° and the maximum rolling angle is 10°. Hence, superhydrophobic coatings are highly promising for self-cleaning [2], drag reduction [3,4], icing resistance [5], antifouling [6,7], antifogging [8], and microfluidic control [9–12]. Furthermore, studies have shown that the superhydrophobic feature of a lotus leaf (a typical example in nature) is due to the microscopic coarse structure and low-surface-energy chemicals on its surface [13,14]. Therefore, the key to artificially fabricating a superhydrophobic surface is to efficiently construct coarse structures with a low surface energy [15,16]. With the development of micro–nano processing technology, many superhydrophobic surface fabrication technologies have been developed, including the sol–gel [17,18], layer-by-layer self-assembly [19–21], etching [22,23], electrochemical deposition [24–26], template [27], and spraying methods [28]. In the liquid spraying method, a solvent-soluble adhesive and nanoparticles are sprayed on the base surface, synchronously or in steps, and dried, forming a superhydrophobic coating. Additionally, this method is very attractive since it is simple and has low equipment and base material requirements, making it available for the large-scale production of superhydrophobic coatings at a low cost [29,30].

In the liquid spraying method, nanoparticles modified with low-surface-energy substances are used as the framework and bound to the adhesive on the base surface, forming

a superhydrophobic coating. Studies on the liquid spraying method can be roughly divided into two categories. The first category is the one-step spraying method, in which appropriate amounts of an adhesive and nanoparticles are jointly dissolved in a solvent, mixed, and sprayed on the surface of the base material. Finally, the surface is dried at an ambient or high temperature to form a superhydrophobic coating [31–36]. Studies have shown that superhydrophobic coatings with an apparent contact angle of 160° could be fabricated using this method, provided the adhesive is compatible with the solvent, the nanoparticles are modified by low-surface-energy substances and have appropriate sizes, and the nanoparticle-to-adhesive ratio is appropriate [37,38]. However, such coatings are prone to lose their superhydrophobicity due to abrasion. Zhang et al. [39] employed the one-step method to fabricate superhydrophobic coatings, which lost their superhydrophobic characteristic after abrasion with 240-grit sandpaper over 0.6 m under a load of 1.6 kPa. The coating prepared by Zhi et al. [40] using the one-step method showed a continuous decrease in the contact angle with increasing friction distance when loaded with 300 g weights and sanded with 600-grit sandpaper. However, none of these studies analyzed the reasons why the coatings were not wear resistant. The superhydrophobic coating prepared by Wei et al. [41] also lost its superhydrophobic property after 20 cycles of abrasion (Taber abrasion test, 250 g load, ASTM D 4060). They believed that after the rough surface structure has been damaged by the sandpaper, the inner layer becomes a flat surface with resin as the main component, which originates from the nanoparticles embedded in the adhesive. Therefore, PDMS [42], cellulose [43], and other additives were added to the solution to improve the wear resistance of such a coating. The results demonstrated that adding appropriate amounts of additives helped to enhance the wear resistance of the coating. However, the superhydrophobicity of the coating decreased and was even lost after being abraded over a longer distance. Therefore, the problem was not solved.

The second category is the two-step spraying method, where the adhesive and nanoparticles are dissolved in a solvent separately. First, the adhesive solution is sprayed on the base material surface. Then, the nanoparticle solution is sprayed on the semi-dried adhesive surface. Finally, the surface is dried at an ambient or high temperature to form a superhydrophobic coating [44–49]. Studies have shown that such coatings are highly superhydrophobic, where the apparent contact angles of liquid droplets could reach up to 160° [50]. Additionally, the wear resistance of such coatings can be strengthened using a sacrificial layer strategy. In other words, the exposed layer is still superhydrophobic after the surface layer of the coating is worn out under an external force. However, the coating may wear off completely, losing its superhydrophobicity when the external force reaches a certain extent. Wang et al. [51] employed the two-step method to fabricate superhydrophobic coatings, which lost their superhydrophobic characteristic after abrasion with 500-grit sandpaper over 2 m under a load of 5 kPa. Therefore, the wear resistance of the coatings fabricated with such a method was limited.

In summary, superhydrophobic coatings fabricated with the existing liquid spraying methods are not wear resistant, and the failure mechanism is unclear. To address this challenge, we analyzed the defects of existing liquid spraying methods in the preparation of wear-resistant superhydrophobic coatings and propose a new method to solve the problem. Superhydrophobic coatings were first prepared using two existing spraying methods and tested for wear resistance via sandpaper abrasion. The failure mechanism of the coatings was investigated through the changes in the apparent contact angle of the droplets and the morphology of the coating surface before and after abrasion. In the one-step method, apart from a few nanoparticles concentrated in the surface layer, the remaining nanoparticles were tightly bound to the interior by the adhesive. After abrasion with sandpaper, the rough structure built up by the surface nanoparticles was destroyed, which then exposed the flatter inner surface where the adhesive predominated, and the coatings lost their superhydrophobic properties. In the two-step method, there was almost no binding force between the adhesive and the coarse structure formed by the nanoparticles. Consequently, such coatings easily fell off under external force and were not wear resistant.

To overcome the shortcomings of existing methods, this paper used phase separation to change the adhesive in the coatings from a continuous-phase adhesive to a discrete-phase adhesive, which was then mixed with nanoparticles and sprayed. This new method allows the nanoparticles to bond to the adhesive while avoiding embedding. Compared to the two existing methods, the superhydrophobic coating prepared using this method retained its excellent superhydrophobic properties even after being subjected to a long abrasion distance using sandpaper. The new method solves the problem of the low wear resistance of coatings obtained using the liquid spraying method and provides a reliable way to realize their practical applications.

## 2. Experiment

### 2.1. Materials and Reagents

The solvents, namely, butyl acetate and ethyl alcohol with a purity of 99.5%, were purchased from Energy Chemical. The adhesive, fluoro-siloxane resin, was purchased from Fuxin Ruifeng Fluorine Chemicals Co., Ltd. (Fuxin, China). Silicon dioxide nanoparticles with a particle size of 20 nm were purchased from Hubei Huifu Nanomaterial Co., Ltd. (Yichang, China). The nanoparticle modifiers, namely, 1H,1H,2H,2H-perfluorodecyltriethoxysilane (PFDTES) with 97% purity, ammonia, and tetraethyl orthosilicate (TEOS), were purchased from Meryer Biochemical Technology Co., Ltd. (Shanghai, China). All the base materials were standard tinplates.

### 2.2. Modifying the Silicon Dioxide Nanoparticles

First, 20 g of nanoparticles was dissolved in a 1 L solution of ethyl alcohol and ammonia (23:2 by volume). Second, the mixture was stirred and processed ultrasonically for 30 min. Later, PFDTES and 3 mL of TEOS were added to the mixture. Third, the mixture was stirred at an ambient temperature for 2 h and thrice cleaned with butyl acetate. Finally, the mixture was centrifuged and dried via heating to obtain fluorinated silicon dioxide nanoparticles.

### 2.3. Existing Preparation Methods

#### 2.3.1. The One-Step Method

First, 7 g of fluoro-siloxane resin was mixed in appropriate amounts of butyl acetate solution, ultrasonically processed for 10 min, and stirred evenly. Second, the mixed solution was added to 2.6 g of fluorinated nanoparticles, ultrasonically processed for 10 min, and magnetically stirred at an ambient temperature for 2 h. Third, the fabricated solution was poured into a spray gun, and the pressure was adjusted to approximately 0.5 MPa. The spray gun was kept vertical, at approximately 20 cm from the tinplate. Finally, the tinplate surface, which was cleaned and sprayed with the solution, was placed in an oven at 120 °C for 2 h to obtain a superhydrophobic coating. The fabrication process is shown in Figure 1a.

#### 2.3.2. The Two-Step Method

First, 7 g of fluoro-siloxane resin was mixed with appropriate amounts of butyl acetate solvent, ultrasonically processed for 10 min, and magnetically stirred for 30 min. Simultaneously, 2.6 g of fluorinated nanoparticles was dissolved in 42 mL of butyl acetate solvent, ultrasonically processed for 10 min, and stirred for 2 h. Second, the fabricated solution was poured into a spray gun and sprayed on a cleaned tinplate under the same spraying conditions as those used in the one-step method. Third, the tinplate was placed in an oven at 80 °C for 10 min and taken out. Fourth, the fluorinated nanoparticle solution was poured into a spray gun and sprayed on the processed tinplate surface under the same spraying conditions. Finally, the tinplate was placed in an oven at 120 °C for 2 h to obtain a superhydrophobic coating. The fabrication process is shown in Figure 1b.

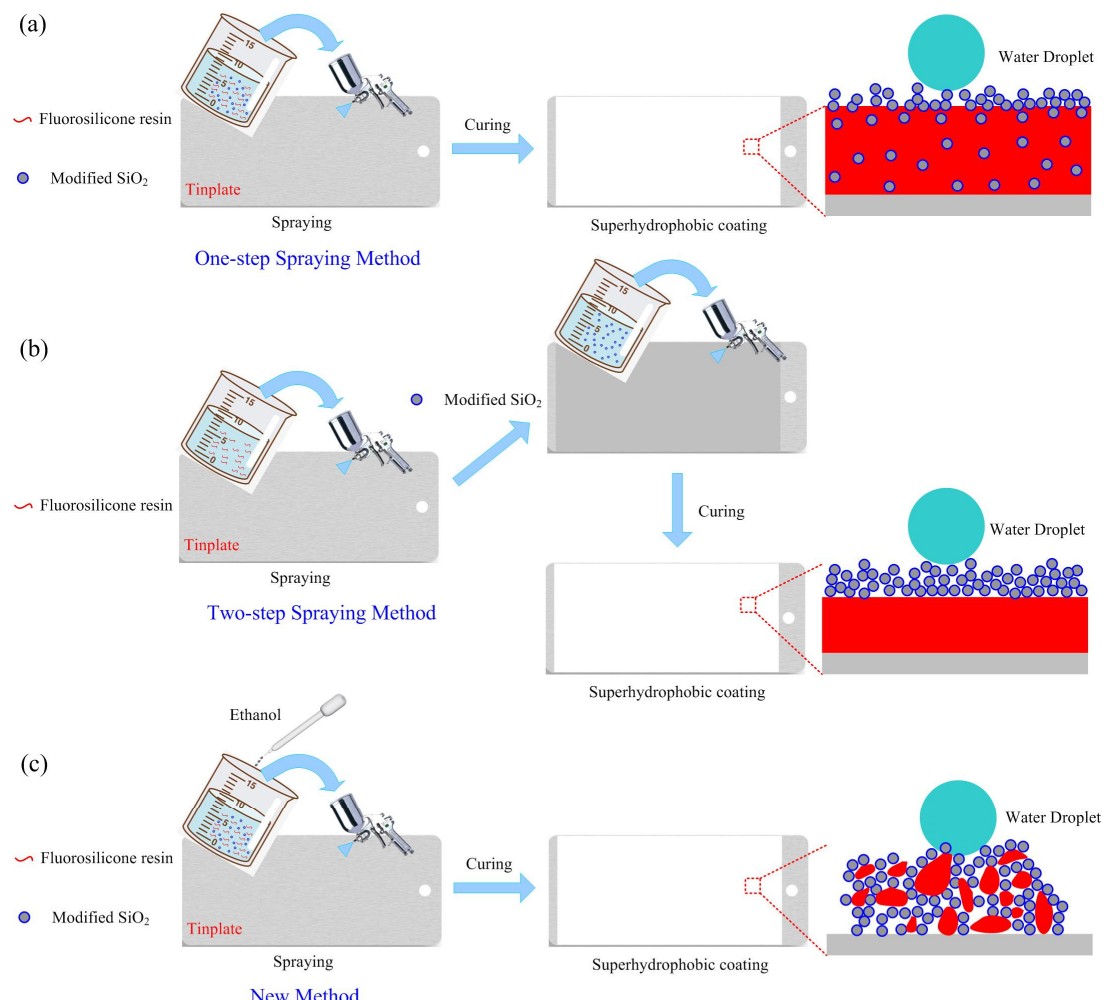

**Figure 1.** Fabrication process of superhydrophobic coatings using different methods: (**a**) the one-step method; (**b**) the two-step method; (**c**) the discrete adhesive method.

### 2.4. A New Route—The Discrete Adhesive Method

First, 7 g of fluoro-siloxane resin was mixed in appropriate amounts of butyl acetate solution, ultrasonically processed for 10 min, and magnetically stirred for 30 min. Additionally, ethyl alcohol was dropped in slowly. Butyl acetate and ethyl alcohol were used in a ratio of 1:3.5 by volume in the solution. Furthermore, phase dispersion occurred in the dropping process since the fluoro-siloxane resin was able to dissolve in butyl acetate but not in ethyl alcohol. Second, 2.6 g of the fluorinated nanoparticles was added to the mixed solution and magnetically stirred for 2 h. Finally, a cleaned tinplate surface, sprayed with the fabricated solution under the same spraying conditions as in the first fabrication method, was placed in an oven at 120 °C for 2 h to obtain a superhydrophobic coating. The fabrication process is shown in Figure 1c.

### 2.5. Performance Characterization

The apparent contact angle and rolling angle were measured using a contact angle meter (DSA 255, 5 µL deionized water droplet, Kruss, Hamburg, Germany). Their values were calculated by averaging the data measured at five positions on the same sample. The apparent appearance of the sample was scanned using a confocal microscope (Olympus, Tokyo, Japan). The wear resistance of the sample was tested with 1000-mesh abrasive paper. The sandpaper was placed on the surface of the 2 cm × 2 cm sample, loaded with a 200 g

weight (pressure of 4.9 kPa), and pulled at a constant speed to abrade the surface of the sample, with a cycle of 10 cm.

## 3. Results and Discussion

Amounts of 7 g of resin and 2.6 g of nanoparticles were used to fabricate coatings with the existing and new methods to facilitate a comparison between the different methods. The mass of nanoparticles used needs to be in the right ratio to the adhesive. If it is too low, the superhydrophobic effect will be lost. For example, the apparent contact angle was less than 150° when using 2.2 g of nanoparticles (Figure 2a). If it is too high, the adhesion between the coating and the substrate will be drastically reduced, and the coating will be easily damaged. Here, 2.6 g of nanoparticles was selected to obtain coatings that reached a contact angle of approximately $156 \pm 1.8°$ and a rolling angle of $3 \pm 0.7°$, as shown in Figure 2a,b, respectively.

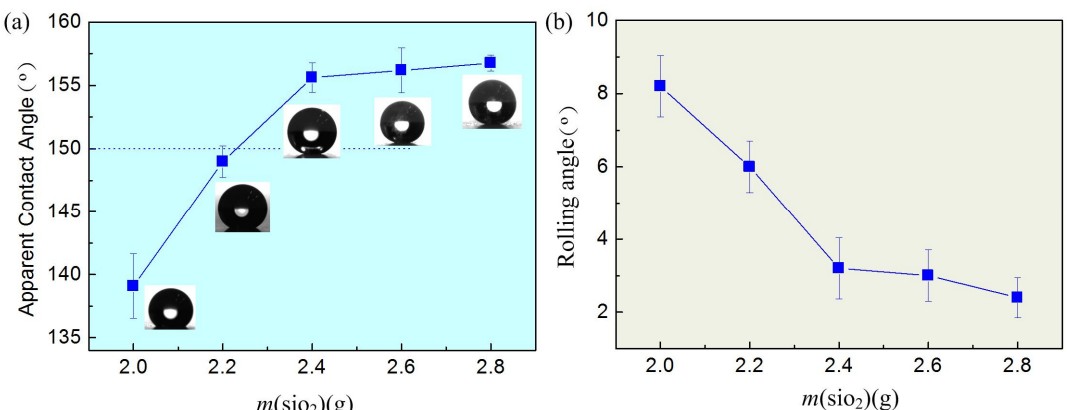

**Figure 2.** The impacts of different nanoparticle masses on the (**a**) apparent contact angle and (**b**) the rolling angle of the coatings.

### 3.1. Analysis of the Failure Mechanism of the One-Step Method

Some works using the one-step method for the preparation of superhydrophobic coatings have yielded poor wear resistance results, despite employing different types of resins, solvents, and nanoparticles. Zhang et al. [39] used a polymer adhesive (PS), microparticles (PP powder), nanoparticles (R812S), and BAC solvent to fabricate superhydrophobic coatings using the one-step method, which lost their superhydrophobic characteristic after abrasion with 240-grit sandpaper over 0.6 m under a load of 1.6 kPa. Zhi et al. [40] employed 1.8 g of epoxy resin, 0.54 g of a curing agent, 0.54 g of hydrophobic silica nanoparticles, and 15 g of acetone to produce a uniform suspension. However, the contact angle on the superhydrophobic coating surface showed a continuous decrease with increasing friction distance. None of these studies explained the failure mechanism.

The same results were obtained for the superhydrophobic coatings prepared in this study. As shown in Figure 3a, the apparent contact angle on the sample surface decreased with increasing abrasion distance. The apparent contact angle decreased from approximately $156 \pm 1.7°$ at the beginning to approximately $146 \pm 1.5°$ after the surface was subjected to an abrasion distance of 3 m, which shows that the sample lost its superhydrophobic characteristic. The contact angle was finally around $110 \pm 1.5°$ after the surface was subjected to an abrasion distance of 7 m.

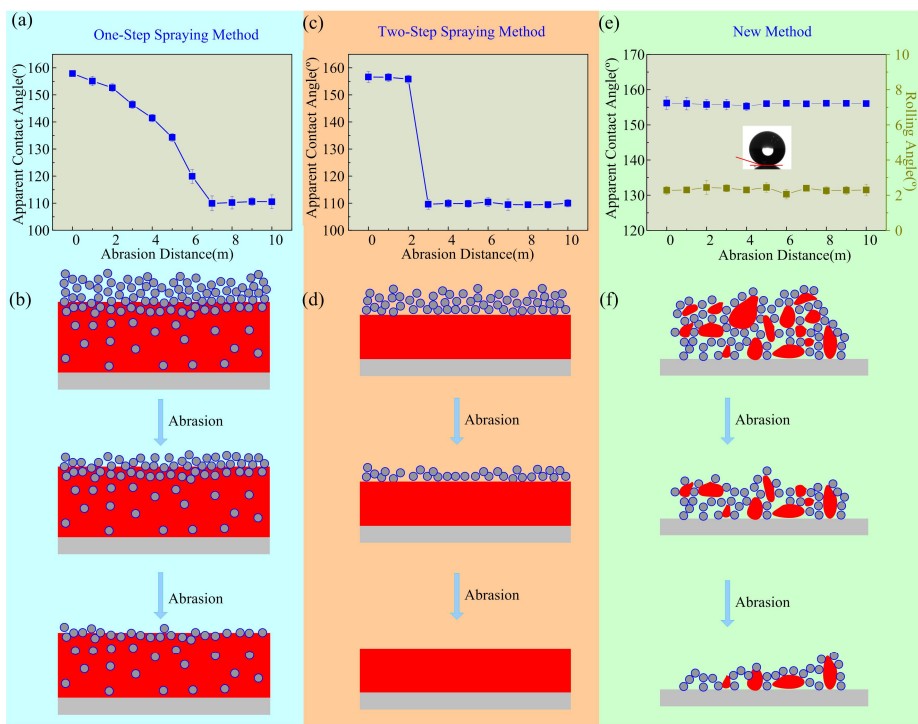

**Figure 3.** Wear resistance test of coatings prepared using different methods. (**a**) The relationship between the apparent contact angle and wear distance in the one-step method. (**b**) Schematic diagram of the wear process in the one-step method. (**c**) Variation in the apparent contact angle during abrasion in the two-step method. (**d**) Schematic diagram of the wear process in the two-step method. (**e**) Variation in the apparent contact angle and rolling angle during abrasion in the new method. (**f**) Schematic diagram of the wear process in the new method.

Wei et al. [41] adopted a silicone-modified polyester (SP) adhesive and silica nanoparticles decorated with perfluorodecyl polysiloxane to prepare a superhydrophobic coating, which also lost its superhydrophobic property after 20 cycles of abrasion. They suggested that the nanoparticles dominated the coating surface at the beginning. The rough structure constructed by the nanoparticles provided the coating with good superhydrophobic properties. As the sandpaper abraded the surface, the rough structure was gradually destroyed. The interior of the coating was dominated by a continuous-phase resin, with a small number of nanoparticles embedded in the resin, which were tightly bound. Once the rough surface structure was destroyed, a relatively flat resin surface was exposed.

In this work, the apparent appearances of the coatings before and after being abraded were compared to find the reason as to why they were not wear resistant. As shown in Figure 4a,b, the confocal microscopy images and typical roughness profiles traced across the coating's surface suggest it had a coarse surface before being abraded. However, the coarse structure on the surface was seriously damaged after being abraded over 3 m, resulting in the sample losing its superhydrophobicity, as shown in Figure 4c,d. Finally, the coating surface almost became flat after being abraded over 7 m, as shown in Figure 4e,f. To confirm the component of the flat surface, the authors dissolved 7 g of fluoro-siloxane resin in certain amounts of ethyl acetate, stirred the mixture for 30 min, and sprayed it on the surface of the base material. Finally, the base material was dried at 120 °C for 2 h. The test results showed that the contact angle of a liquid droplet on the surface was almost 110°, as shown in Figure 4f. Therefore, the data prove that the resin occupied the vast majority of the coating surface after being abraded over 7 m. Our test results totally agree with those of Wei et al. [41]. The entire wear process of the coating is shown in Figure 3b.

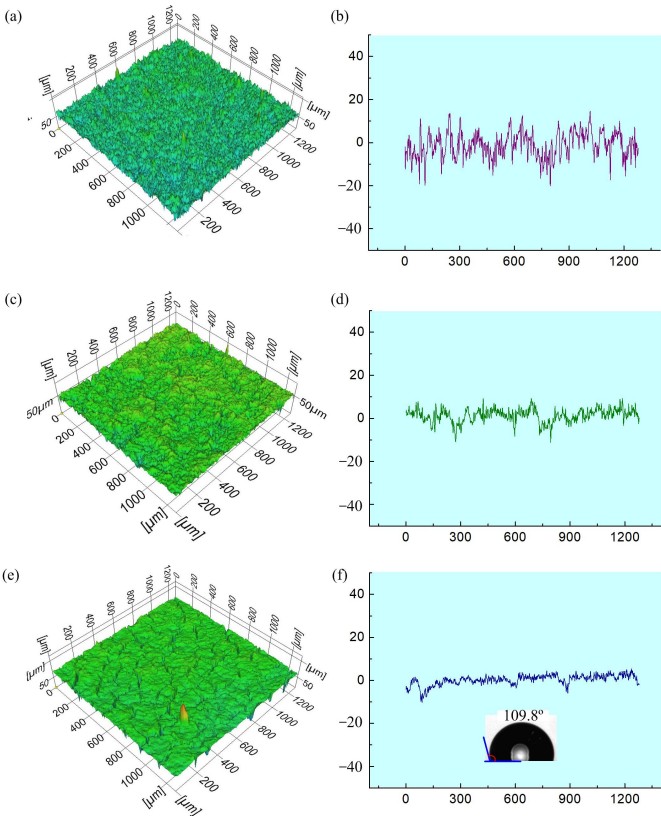

**Figure 4.** Wear resistance test of a coating fabricated with the one-step method. Confocal microscopy images and corresponding roughness profiles of the coating (**a**,**b**) before being abraded, (**c**,**d**) after being abraded over 3 m, and (**e**,**f**) after being abraded over 7 m. All the dimensions are in μm.

In combination with the above experimental results, and in comparison with the work of Wei et al. [41], we suggest that the reason for the lack of wear resistance of the coatings lies in the overly tight bonding of the resin to the nanoparticles, resulting in a relatively flat surface with a predominantly resinous inner structure, and the subsequent loss of the superhydrophobic properties of the coating when the rough surface structure is destroyed.

### 3.2. Analysis of the Failure Mechanism of the Two-Step Method

The two-step method used by Wang et al. [51] was as follows: First, a layer of hydrocarbon resin adhesive was sprayed on a clean substrate surface. Then, the prepared silica ethanol suspension was sprayed onto the primer.

The sliding angle was around $7.0 \pm 1.9°$ after abrasion with 500-grit sandpaper over 2 m under a load of 5 kPa. However, there was a sharp increase in the sliding angle after the surface was abraded over a distance of 3 m. In this work, as shown in Figure 3c, the apparent contact angle of the coating was maintained at around $156 \pm 1.5°$ over an abrasion distance of not more than 2 m. However, the contact angle sharply decreased to approximately $110 \pm 1.2°$ when the abrasion distance reached 3 m.

The apparent appearances of the sample before and after abrasion were analyzed to explore the cause of the change in the apparent contact angle of the coating. The coating surface was coarse before (Figure 5a,b) and even after being abraded over 2 m (Figure 5c,d). Therefore, over the abrasion distance of 2 m, the coating maintained its superhydrophobicity. However, when the abrasion distance reached 3 m, there was no white coating on the sample surface, even after a visual check, as shown in Figure 5e. Since the fluorinated nanoparticle solution was only sprayed on the surface of the resin adhesive, there was a very weak bonding strength between the coarse structure and the resin adhesive. According to the spraying method and the apparent contact angle of the

coating of approximately 110°, it could be determined that the resin on the coating was exposed after being subjected to an abrasion distance of 3 m. Wang et al. [51] also presented the scanning electron microscopy (SEM) image of the surface after being abraded over a distance of 10 m. The surface was severely damaged, and the multiscale structures disappeared.

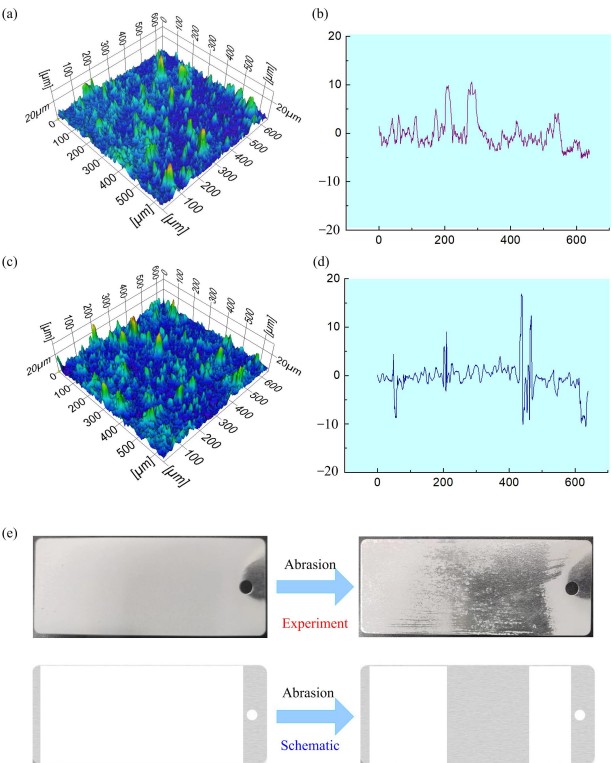

**Figure 5.** Wear resistance test of a coating fabricated with the two-step method. Confocal microscopy images and corresponding roughness profiles of the coating (**a**,**b**) before being abraded and (**c**,**d**) after being abraded over 2 m. (**e**) Experimental and schematic images of the coating after being abraded over 3 m. All the dimensions are in μm.

The entire wear process of the coating is shown in Figure 3d. The whole coating surface was initially composed of coarse structures formed by nanoparticles. However, due to the weak bonding strength between the coarse structure and the resin adhesive, the coarse structure was highly prone to damage by abrasion. The new exposed surface was still composed of a coarse structure formed by nanoparticles, even though the coarse structure on the surface layer was damaged over short abrasion distances, maintaining the superhydrophobicity of the coating. However, the coarse structure was fully worn out when subjected to increased abrasion distances, meaning the coating lost its superhydrophobicity. Therefore, the key reason for the lack of wear resistance in such a coating is that its coarse structure is easy to damage since there is a weak bonding strength between the nanoparticles and the resin adhesive.

### 3.3. The Discrete Adhesive Method to Prepare Wear-Resistant Superhydrophobic Coating

The analyses conducted thus far in this study revealed that, in the coating fabricated with the one-step method, the nanoparticles, except those on the surface layer, were embedded in the adhesive due to the excessive binding between the two. Consequently, the inner layer of the coating was a flat surface mainly occupied by the resin, resulting in the poor wear resistance of the coating [41]. On the contrary, the coarse structure formed by the nanoparticles only covered the resin adhesive without any binding force, resulting in an easy fall-off and lower wear resistance for the coating fabricated with the two-step

method. Therefore, excessive and insufficient binding between the resin and nanoparticles is not conducive to enhancing the wear resistance of the coating. Obtaining the proper binding degree between the two is the key to improving the wear resistance of the coating.

In order to realize the above principle, we first used phase separation to change the fluorosilicone resin from a continuous phase to a discrete phase [52], and then mixed it with the nanoparticles for spraying. The purpose was to avoid the defects in the coatings fabricated with the one-step and two-step methods. The resin could be dispersed through phase dispersion so that the nanoparticles could bind with the resin without being embedded in it, realizing an appropriate binding of the two components.

We first characterized the static and dynamic superhydrophobic properties of the coating prepared using the new method. It can be seen from Figure 3e that the apparent contact angle of the coating was as high as $156 \pm 0.6°$, and the rolling angle was as low as $2.3 \pm 0.3°$. The water droplets were able to bounce on the coating in a short time under the condition of a Weber number of 9.6 (droplet radius was 1.5 mm, Figure 6a). These results prove that the coating has excellent static and dynamic superhydrophobicity.

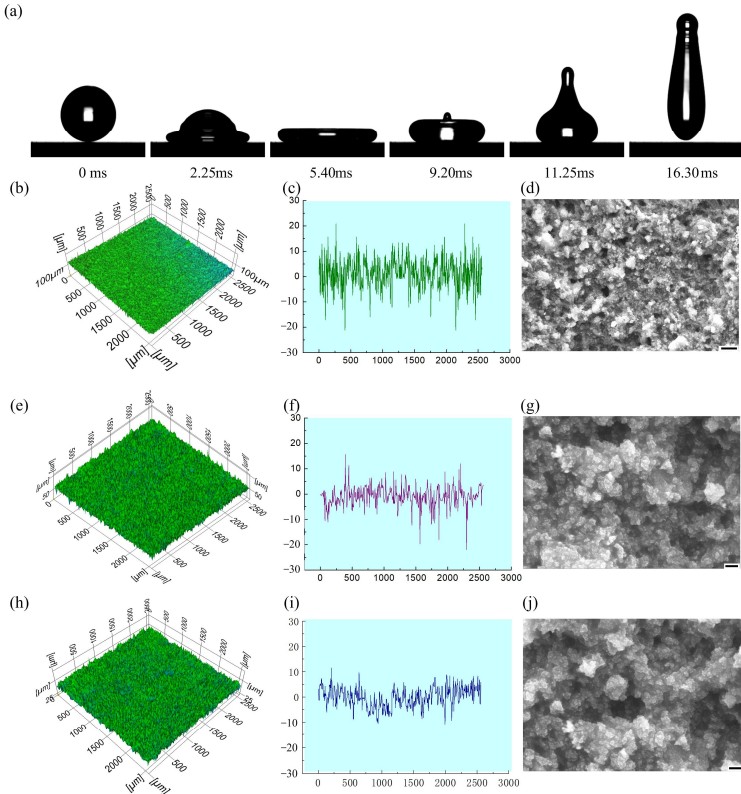

**Figure 6.** (**a**) The impact of water droplets on the coating fabricated with the discrete adhesive method. Confocal microscopy images, corresponding roughness profiles, and SEM images of the coating (**b–d**) after being abraded over 3 m, (**e–g**) after being abraded over 7 m, and (**h–j**) after being abraded over 10 m. All the dimensions are in μm. The scale bars of (**d,g,j**) are 4, 1, and 1 μm, respectively.

Furthermore, the wear resistance of the superhydrophobic coating fabricated with the new method was tested. The test results are shown in Figure 3e. The apparent contact angle and the sliding angle of a liquid droplet on the coating were consistently maintained at around $156 \pm 0.6°$ and $2.3 \pm 0.3°$, even over an abrasion distance of 10 m. Comparing the results in Figure 3a,c, we can clearly see the advantages of the new method in terms of the improvement in wear resistance.

We characterized the surface morphology of the coating to explain the reason for its excellent wear resistance. In this paper, the surface morphology of the coating after being subjected to wear distances of 3 m (Figure 6b–d), 7 m (Figure 6e–g), and 10 m (Figure 6h–j)



was characterized using confocal microscopy images, corresponding roughness profiles, and SEM images. It can be seen that, for each wear distance, the coating surface always contained numerous rough structures, ensuring its excellent superhydrophobic properties.

The entire wear process of the coating is shown in Figure 3f. From the results in Figure 6b–j, it can be inferred that the coating prepared using this method formed a self-similar structure from the outside to the inside [41]. Even if the surface layer falls off, the inner layer will still have numerous rough structures, which can ensure its excellent superhydrophobic properties.

## 4. Conclusions

The preparation of wear-resistant superhydrophobic coatings using large-scale fabrication methods has always been one of the challenges that limits their progress towards practical applications. In this work, we analyzed two existing liquid spraying methods that can be used to produce superhydrophobic coatings at a large scale and pointed out that the reason for the difficulty in producing wear-resistant coatings is that the degree of binding between the adhesive and the nanoparticles is either too high or too low. Therefore, a new method was proposed to transform the adhesive from a continuous-phase adhesive to a discrete-phase adhesive, which can sufficiently achieve the bonding between the adhesive and the nanoparticles and solve the problem of poor wear resistance. In addition to their wear resistance, other properties of coatings prepared using this method, such as adhesion, hardness, anti-wetting properties, acid and alkali resistance, and aging resistance, can be investigated in the future in order to further optimize the quality or type of resin and nanoparticles used in the method. This may help to eventually realize the application of the new method in actual practice.

**Author Contributions:** Writing-original draft, X.L.; Supervision, X.L., C.X. and L.W.; Project administration, X.L.; Funding acquisition, X.L.; Validation, Z.S. and L.L.; Investigation, X.S.; Formal analysis, J.W.; Writing-review and editing, Z.S., L.L., X.S., J.W., C.X. and L.W. All authors have read and agreed to the published version of the manuscript.

**Funding:** The authors acknowledge the support from the National Science Foundation of China (Grant No. 22203038), the Education Department of Hebei Province (QN2020255), the Doctor Scientific Research Start-up Costs of Langfang Normal University (XBQ202027), and the Key Research and Development Projects of Hebei Province (19227133D).

**Institutional Review Board Statement:** Not applicable.

**Informed Consent Statement:** Not applicable.

**Data Availability Statement:** All data that support the findings of this study are included within the article.

**Conflicts of Interest:** The authors declare no conflict of interest.

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
