# Peer review of "Preparation of Wear-Resistant Superhydrophobic Coatings Based on a Discrete-Phase Adhesive"

_coatings, doi:10.3390/coatings13040682_

Round 1

Reviewer 1 Report

Interesting paper on the analysis of a new method for the application of coatings. It fits on the scope of the paper, but some major revisions are required before it is accepted.

English should be improved. The paper needs to be rewritten to correct verb tenses and improve text fluency.

Lines 14-15: Sentence is unclear. You might want to revise it to something about the unpredictability of wear-resistance properties on the common coating preparation methods. The names of the methods should be mentioned.

Introduction: Consider adding more references regarding the existing two liquid spraying methods.

Line 32: “microcosmic”? you mean “microscopic”?

Line 165 : “The same masses of resin” quantify.

Line 168 – 170 : Why 2.6g of nanoparticles ? The reasons should be mentioned.

Section 3.1 - Analysis of the failure mechanism of the one-step method (Line 175): Before starting to mention the literature, write an introductory sentence to introduce such discussion.

Figure 3 – Axis titles and scales should be increased. Its difficult to read. The caption is confusing.

There are two Figures named “Figure 4”, please revise. Also change a1), a2),.. to a), b), c) ….and so on. Describe what they are in the caption. Also, there is no need to present [μm] in every axis, you can just write: “All dimensions are in μm.” Additionally, In the first Figure 4, what is the purpose of presenting figures a2), b2) and c2)? From where do they come from? Their purpose should be mentioned in the text. Same for the second Figure 4.

Line 251: “SEM”. The first time an abbreviation is presented it should be like: “ Scanning electron microscopy (SEM)”.

Figure 5: Same problem as Figure 4. Also, the scale in the SEM image are not easily visible. Consider locating them outside the image.

Conclusion: Consider highlighting what can be done in future studies to further advance the methodology proposed in the present work.

Reviewer 2 Report

The article "Preparation of wear-resistant superhydrophobic coatings based on adhesive with a discrete phase" is devoted to the creation of superhydrophobic coatings. This topic is of great interest in various industries. The results obtained are of particular interest to a wide range of readers. The materials of the article are presented in an understandable form and well structured. References are relevant. The quality of the figures is high, the process diagrams are clear.

I recommend accepting the article for publication after the following minor corrections:

1. In the Introduction section, it would be advisable to clearly state the purpose (aim) of the work.

2. In conclusion, it would be advisable to indicate specific values ​​based on the results of the work, and not just say discrete method is better.

Round 2

Reviewer 1 Report

The authors replyed to all my comments. The paper has improoved.